# The Use of Tocilizumab in Patients with COVID-19: A Systematic Review, Meta-Analysis and Trial Sequential Analysis of Randomized Controlled Studies

**DOI:** 10.3390/jcm10214935

**Published:** 2021-10-25

**Authors:** Alberto Enrico Maraolo, Anna Crispo, Michela Piezzo, Piergiacomo Di Gennaro, Maria Grazia Vitale, Domenico Mallardo, Luigi Ametrano, Egidio Celentano, Arturo Cuomo, Paolo A. Ascierto, Marco Cascella

**Affiliations:** 1First Division of Infectious Diseases, Cotugno Hospital, AORN dei Colli, 80131 Naples, Italy; albertomaraolo@mail.com; 2Epidemiology and Biostatistics Unit, Istituto Nazionale Tumori, IRCCS Fondazione G. Pascale, 80131 Naples, Italy; piergiacomo.digennaro@live.it (P.D.G.); e.celentano@istitutotumori.na.it (E.C.); 3Department of Public Health and Infectious Diseases, Sapienza University of Rome, Viale del Policlinico 155, 00161 Rome, Italy; m.piezzo@breastunit.org; 4Department of Melanoma, Cancer Immunotherapy and Development Therapeutics, Istituto NazionaleTumori, IRCCS Fondazione G. Pascale, 80131 Naples, Italy; dott.mariagrazia.vitale@gmail.com (M.G.V.); dome.mallardo@gmail.com (D.M.); paolo.ascierto@gmail.com (P.A.A.); 5Department of Clinical Medicine and Surgery, Section of Infectious Diseases, University of Naples Federico II, 80131 Naples, Italy; luigi.ametrano@outlook.com; 6Division of Anesthesia and Pain Medicine, Istituto Nazionale Tumori, IRCCS Fondazione G. Pascale, 80131 Naples, Italy; a.cuomo@istitutotumori.na.it (A.C.); m.cascella@istitutotumori.na.it (M.C.)

**Keywords:** COVID-19 pneumonia, tocilizumab, SARS-CoV-2, COVID-19, meta-analysis, trial sequential analysis

## Abstract

Background: Among the several therapeutic options assessed for the treatment of coronavirus disease 2019 (COVID-19), tocilizumab (TCZ), an antagonist of the interleukine-6 receptor, has emerged as a promising therapeutic choice, especially for the severe form of the disease. Proper synthesis of the available randomized clinical trials (RCTs) is needed to inform clinical practice. Methods: A systematic review with a meta-analysis of RCTs investigating the efficacy of TCZ in COVID-19 patients was conducted. PubMed, EMBASE, and the Cochrane COVID-19 Study Register were searched up until 30 April 2021. Results: The database search yielded 2885 records; 11 studies were considered eligible for full-text review, and nine met the inclusion criteria. Overall, 3358 patients composed the TCZ arm, and 3131 the comparator group. The main outcome was all-cause mortality at 28–30 days. Subgroup analyses according to trials’ and patients’ features were performed. A trial sequential analysis (TSA) was also carried out to minimize type I and type II errors. According to the fixed-effect model approach, TCZ was associated with a better survival odds ratio (OR) (0.84; 95% confidence interval (CI): 0.75–0.94; I^2^: 24% (low heterogeneity)). The result was consistent in the subgroup of severe disease (OR: 0.83; 95% CI: 0.74–0.93; I^2^: 53% (moderate heterogeneity)). However, the TSA illustrated that the required information size was not met unless the study that was the major source of heterogeneity was omitted. Conclusions: TCZ may represent an important weapon against severe COVID-19. Further studies are needed to consolidate this finding.

## 1. Introduction

Tocilizumab (TCZ) is a humanized monoclonal antibody that, via the binding to soluble and membrane interleukin (IL)-6 receptors, produces inhibition of the proinflammatory signals [1]. It is commonly used in several types of inflammatory arthritis, in Castleman’s syndrome, and in cytokine release syndrome secondary to chimeric antigen receptor T cell therapies [2]. Given its ability to intercept proinflammatory cascades, TCZ is potentially useful in all clinical conditions produced by the dysregulation of inflammatory processes, especially when refractory to other approved treatments [3].

Although the precise pathogenesis of the coronavirus disease 2019 (COVID-19) pneumonia remains unsolved, evidence showed that within a complex cytokine storm scenario, SARS-CoV-2 provokes a dramatic increase in IL-6 levels [4]. Based on this evidence, it was suggested to use TCZ for improving the patients’ outcomes in COVID-19 pneumonia [5]. Consequently, many clinical studies have been conducted to evaluate the efficacy of this treatment, and an increasing number of evidence-based medicine analyses can be found in the literature [6].

However, initial evidence syntheses failed to produce definitive results, especially owing to the conflicting findings emerging between observational studies and randomized clinical trials (RCTs) [7]. As matter of fact, in evaluating the effectiveness of drugs for the treatment of COVID-19, even in high-impact journals, the following methodological distortions were common among observational studies, particularly dealing with time-to-event analysis: immortal time bias, confounding bias, and competing risk bias [8].

Despite the rush for a game-changing treatment capable of significantly impacting the prognosis of COVID-19 patients, clinical practice must rely upon rock-solid evidence, and well-known RCTs are placed on the top of the hierarchy of evidence [9]. This is the rationale behind the choice to perform a systematic review and meta-analysis only focused on RCTs, involving the comparison of TCZ with placebo or standard of care (SoC), for the treatment of COVID-19.

## 2. Materials and Methods

This systematic review is PROSPERO registered (registration number: CRD42020226657) and complies with the Preferred Reporting Items for Systematic Reviews and Meta-Analyses (PRISMA) statement in its 2020 version (PRISMA Statement, Ottawa, ON, Canada) [10].

### 2.1. Search Strategy

PubMed, EMBASE, and the Cochrane COVID-19 Study Register were searched up until 30 April 2021 for RCTs to investigate the efficacy of TCZ in COVID-19 patients. The search was restricted to peer-reviewed articles. Neither geographical nor language restrictions were applied. The search strategies were designed by two researchers in the team (A.E.M., L.A.), using appropriate combinations of the following keywords through Boolean operators: “tocilizumab”, “COVID-19”, “SARS-CoV-2”, “2019-nCoV”, “novel coronavirus”. Strategies for retrieving articles were adapted according to the databases’ distinctive features. Specific details are provided in Appendix A. Manual checking of reference lists and citation tracking of included papers were undertaken in order to retrieve further articles.

### 2.2. Screening and Eligibility

Duplicate records were discarded by using the EndNote 20 reference managing software (Clarivate, Philadelphia, PA, USA) [11]. A two-step screening process for eligibility was carried out. First, two authors (M.D.G. and D.M.) excluded ineligible studies by screening titles and abstracts. Then, another couple of authors (A.C., M.P.) independently reviewed the full texts of potentially eligible studies for inclusion in the review. Disagreements were resolved through discussion and general consensus. Eligibility was assessed by resorting to the PICOS (population, intervention, comparators/controls, outcomes and study design) question format [11,12], as follows:Population: Patients affected by COVID-19.Intervention: Administration of tocilizumab, alone or in association with other drugs.Comparators/controls: SoC, placebo, or any kind of alternative interventions.Outcomes: The main outcome of interest was all-cause mortality reported using an intention-to-treat (not modified) method (28-/30-day mortality, in-hospital mortality, overall survival according to the reported results). Secondary outcomes were represented by clinical success, time to recovery, rate of intensive care unit (ICU) admission, risk of mechanical ventilation requirement, duration of mechanical ventilation, length of stay (LOS), safety profile related to pharmacological intervention, toxicity, rate of secondary infection, and time to hospital discharge.Study design: Only RCTs were included.

Studies that did not fulfill the eligibility criteria were excluded. The minimum sample size required was at least 50 patients per arm.

### 2.3. Data Extraction

Two authors (A.C. and M.P.) independently abstracted data from each study and the data were subsequently double entered into a custom-made electronic database (an Excel spreadsheet) to eliminate data entry errors. Discrepancies were resolved by consensus, or with a third author (A.E.M.) if necessary. Data compilation used a standardized data extraction tool to report the following variables of interest: first author; country; sample size; main population features (e.g., age, gender, comorbidities); criteria of COVID-19 diagnosis; COVID-19 severity; setting (outpatient/in-patient, non-ICU/ICU); intervention characteristics (TCZ schedule, companion agent if present); comparator features; survival outcome measures such as number of deaths, odds ratios (ORs), and hazard ratios (HRs); and follow-up duration. Data regarding the main outcome were extracted for the whole study population and major subgroups of interest. When necessary, graphical data abstraction was conducted using open-source software. In addition, the full protocol of each study was consulted to verify the study objectives, population, and other relevant information regarding the study design and conduction.

### 2.4. Data Synthesis and Analysis

To obtain more appropriate estimates of the average treatment effect in the case of between-study heterogeneity, the pooled estimates of ORs with two-sided 95% CIs were computed for 28-day mortality. For this aim, a fixed-effect model according to the inverse-variance method [13], and the random-effect model of DerSimonian and Laird [14], were adopted. The assumption of homogeneity between studies was tested with Cochran’s Q statistics, and the measure of the degree of inconsistency across studies was assessed with Higgins’ I^2^ index, quantifying heterogeneity as low, moderate, and high, with upper limits of 25, 50, and 75% for the I^2^ values, respectively [15]. Predefined subgroups were analyzed to better understand if the treatment effect changed because of specific trial and patients characteristics. All results were displayed by specific forest plots. A *p*-value < 0.05 was considered statistically significant. Sensitivity analysis was carried out according to the leave-one-out cross validation method that calculates the pooled estimates omitting one study at a time, to capture some features of the included studies that are able to influence the pooled estimates. Funnel plots and regression tests, according to the method reported by Egger [16], were performed to assess the publication bias. Data analysis was performed using R 3.4.1 software packages (The R Foundation for Statistical Computing, Wien, Austria) [17,18]. The summary statistics to measure treatment effect were represented by the OR and presented along with appropriate 95% CI values.

Trial sequential analysis (TSA) was also performed for better interpreting the meta-analysis results, since it can minimize the risk of making a falsely positive/negative conclusion, thereby producing more conservative thresholds for statistical significance [19]. TSA combines conventional meta-analysis methodology with repeating significance testing methods applied to accumulating data in clinical trials. It calculates cumulative z-curves and uses the law of the iterated logarithm to penalize the Z value and to produce more conservative meta-analysis results [20]. TSA was performed using TSA software, version 0.9.5.10 Beta (Copenhagen trial unit, https://ctu.dk/tsa/, (accessed on 28 May 2021)).

### 2.5. Quality Appraisal

The Cochrane Collaboration tool for assessing risk of bias in RCTs was implemented to gauge the quality of the included studies [21]. The following items were evaluated: random sequence generation; allocation concealment; blinding of participants and personnel; blinding of outcome assessment; incomplete outcome data; selective reporting; and other potential sources of bias [21]. Risk of bias for each study was independently assessed by M.P. and P.D.G., and disagreements were discussed and resolved by consensus between both reviewers or by consultation with a third reviewer (A.C.).

## 3. Results

### 3.1. Study Selection and Characteristics

After de-duplication from an initial total of 2885 records, the titles and abstracts of 1589 studies were screened. Overall, 11 studies were considered eligible for full-text review, and nine met the inclusion criteria [22,23,24,25,26,27,28,29,30]. Figure 1 depicts the entire process of study identification, inclusion, and exclusion. Details of the included studies are available in Table 1. Overall, nine trials were included, enrolling 3358 patients in the TCZ group and 3131 subjects in the comparator group. Studies were conducted from March 2020 to early 2021 across several countries worldwide; all trials were multicenter. The enrolled patients suffered from moderate to critical disease, according to the definitions provided by the United States National Institutes of Health (NIH), as far as the clinical spectrum of SARS-CoV-2 infection is concerned [31]. Mortality was not always the primary endpoint but was assessed in all trials at 28 or 30 days except in one study, where researchers investigated in-hospital mortality [23]. Tocilizumab dosing was quite variable, ranging from 6 to 8 mg/kg, administered as a single dose or repeated short term.

### 3.2. Risk of Bias

Since the majority of trials (6 out of 9) were open-label studies, performance bias and detection bias were classified as “high”. The remaining components of risk, such as selection, attrition, and reporting bias, were classified as “low” for all trials included in this analysis. The risk-of-bias in each study is reported as Appendix A.

### 3.3. Meta-Analysis of Main Outcome

All nine included RCTs concurred with the main analysis. The raw death rate was 24.8% in the TCZ group (835/3358) and 29.9% (935/3131) in the control group. According to the fixed-model approach, TCZ was associated with lower mortality in a statistically significant way (OR: 0.84; 95% CI: 0.75–0.94; I^2^: 24% (low heterogeneity)). The results were consistent when implementing a random-effect model, although a widening of the CI was observed, including the vertical line (OR: 0.87; 95% CI: 0.71–1.07). All results related to the primary analysis are depicted in the forest plot presented in Figure 2.

### 3.4. Subgroup Analysis

Pre-planned key subgroup analyses were carried out to explore how the treatment effect varied across different subsets of studies or patients. When contrasting open label with placebo-controlled trials (Figure 3), the beneficial effect of TCZ on mortality was confirmed in the subgroup that included the first type of studies (OR: 0.82; 95% CI: 0.72–0.92; I^2^: 36%; fixed-effect model), but the benefit disappeared in the other subgroup (OR: 1.12; 95% CI: 0.75–1.66; I^2^: 0%; the results were the same according to fixed- and random-effect models), although not statistically significant. The results obtained when testing for subgroup difference were also not significant, so no interaction existed between the subtotal estimates for the subgroups.

Another subgroup analysis concerned the use of TCZ alone or with the SoC, in addition to the type of comparator: SoC with or without placebo (Figure 4). In the most numerous subgroup of TCZ versus SoC, even if including only three studies [22,25,27], the positive effect of TCZ on mortality was apparent (OR: 0.80; 95% CI: 0.71–0.91; I^2^: 19%; with the fixed-effect model being the result of the overlapping of the random-effect model), in contrast to the other subgroups. The results obtained when testing for subgroup difference were not statistically significant.

The last subgroup analysis involved the spectrum of disease severity, after extracting data on homogeneous categories of patients when data were available (Figure 5). The impact of TCZ in patients with moderate diseases seemed not to be clinically relevant (OR for mortality: 1.30; 95% CI: 0.64–2.64; I^2^: 0%; these values were equal to the results of fixed-effect and random-effect analysis), whereas it appeared beneficial when considering severe/critical disease. Nevertheless, in this case, there was a discrepancy between the fixed-effect model (OR: 0.84; 95% CI: 0.75–0.94; I^2^: 53%) and random-effect model (OR: 0.89; 95% CI: 0.71–1.18). No significant interaction existed between the subgroups when considering treatment effects.

### 3.5. Quality Appraisal and Publication Bias

The results of quality assessment are depicted in Appendix A: the major issues were related to open label studies [22,23,26,28,29,30] due to the lack of blinding. The results of stratifying studies according to this criterion (open label versus placebo-controlled) were already shown in Figure 3. The funnel plots of the primary outcomes of the studies are presented, along with the results of Egger’s regression test (*p* = 0.1441), which suggest an absence of publication bias and small-study effects.

### 3.6. Sensitivity Analysis

In Appendix A, the results of the sensitivity analysis are described. It was performed using the leave-one-out method and shows that the estimated pooled ORs, obtained excluding one study at time, are still consistent, even when omitting the study with the highest weight according to both the fixed-effect (75%) and random-effect 43.4%) models [28] (OR: 0.86; 95% CI: 0.68–1.07). This held true even when omitting the study that was apparently the major driver of heterogeneity, the TOCIBRAS study [30]: when it was excluded, the I^2^ dropped to 0% and the OR for mortality associated with TCZ use was 0.82 (95% CI: 0.74–0.92).

### 3.7. Trial Sequential Analysis

In our trial sequential analysis, the type I error risk was set at α = 0.05 with a power of 0.80. In this condition, the required information size (RIS) for the meta-analyzed estimate was 7786, while our included number was 6489 subjects, even if the cumulative z-curve crossed above 1.96, which corresponded to the nominal threshold for statistical significance, demonstrating that the effect of tocilizumab seems to be more effective in reducing the 28/30-day mortality (Figure 6A). This conclusion was confirmed when omitting the TOCIBRAS trial, which was the major source of heterogeneity (Appendix A), from TSA, since the cumulative z-curve also reached the RIS (Figure 6B).

## 4. Discussion

The results of our systematic review and meta-analyses are in line with the most recent development of the recommendations for COVID-19 treatment, which now include TCZ as an important option for patients with the severe or critical disease [32]. These data need to be put into context to understand how TCZ has become a potential life-saving agent in patients with SARS-CoV-2 infection.

Indeed, until mid-2021, the therapeutic armamentarium for patients with COVID-19 was quite bereft of effective weapons. Steroids, particularly dexamethasone, were the first class of drugs to show benefits in terms of the mortality of patients affected by severe SARS-CoV-2 infection [33]. Antiviral treatments failed to show any relevant impact on overall survival. This also applies to hydroxychloroquine/chloroquine [34], studies of which initially sparked much interest but, with the benefit of hindsight, appear to be biased by serious flaws inherent to the observational nature of these first studies [35], and more importantly to remdesivir, which has no or little effect on mortality [36] but might improve time to recovery and recovery rate [37]. As matter of fact, the window of opportunity for effective antiviral therapy is very narrow and it is open only in the early phase of the disease [38] when viral load peaks [39]. Afterward, the dysregulated hyperinflammatory response dominates in severe cases and is responsible for the most relevant manifestations [40]. Notably, although no single definition is widely accepted, the cytokine storm is an umbrella term encompassing many disorders whose shared hallmark is an immune dysregulation that potentially leads to multiorgan dysfunction [41]. The cytokine storm may be pathogen-induced, as seen in SARS-CoV-2 infection, iatrogenic, as observed in CAR (chimeric antigen receptor) T-cell therapy, or may ensue from autoimmune, neoplastic, or idiopathic causes [41].

Many factors contribute to the pathophysiology of the cytokine storm. Among the many stands is IL-6, whose circulating concentrations are known to be increased in many proinflammatory critical care syndromes, including COVID-19 [42]. It is a master cytokine, produced by—and acting on—immune and non-immune cells in multiple organ systems. IL-6 exerts pleiotropic effects, not only driving inflammation, fever, and carcinogenesis, but also regulating metabolism, bone turnover, and hematopoiesis, and thus, is fundamental for innate and adaptive immunity [42].

In light of its important physiological and anti-inflammatory functions, the blockade of IL-6 signaling might represent a double-edged sword but has turned out to be effective in some cytokine storm disorders, such as idiopathic multicentric Castleman’s disease and CAR T-cell-induced cytokine release syndrome, through monoclonal antibodies that are directed at the IL-6 receptor (TCZ and sarilumab) or directly target IL-6 (siltuximab) [43].

Elevations in serum IL-6 levels in patients affected by severe COVID-19 have spurred a renewed interest in this cytokine as a therapeutic target in the broader context of the cytokine storm syndrome triggered by SARS-CoV-2 infection [5,44]. Despite the logistical difficulties, several studies have been conducted in a short time. Predictably, observational studies in particular suffered from relevant methodological limitations that threatened the validity of their conclusions [45]; moreover, they yielded inconsistent results when pooled with RCTs in evidence syntheses [46].

The need for high-quality data to inform clinical practice led to research efforts focusing only on RCTs. A Cochrane review published in March 2021 retrieved data up to the end of February [47], including eight RCTs investigating TCZ [22,23,24,25,26,27,28,29] and one RCT testing sarilumab [48], with pooling of the available information with respect to mortality. TCZ appeared to be effective in reducing 28-day mortality compared with SoC or placebo (relative risk: 0.89; 95% CI: 0.82 to 0.97; I^2^: 0.0%), but the evidence was less certain when considering 60-day mortality, clinical improvement, and adverse events [47]. Remarkably, the largest trial, the RECOVERY study, was only in pre-print form at that time and its overall results were partially incomplete [28]. Several factors have been suggested to explain the differences in clinical outcomes highlighted by the Cochrane review. They include important differences in trial designs, the features of included patients, stages of disease, the use of co-interventions (e.g., the proportion of concomitant steroids), and the endpoint measurement scales [49].

The full publication of the RECOVERY trial [28] and the accumulation of further evidence regarding TCZ and other IL-6 blockers led to a prospective meta-analysis (on 27 RCTs) of utmost importance in June 2021. It showed that the use of IL-6 antagonists was associated with improved survival in COVID-19 patients, but the results were statistically significant only for TCZ (OR for mortality equal to 0.83; 95% CI: 0.74–0.92) [50]. Coherently with what was observed in the RECOVERY study, the subgroup of patients who received also corticosteroids appeared to benefit the most: the mortality risk was even lower (OR: 0.77; 95% CI: 0.68–0.87) [32]. On this basis, the World Health Organization (WHO) issued a strong recommendation on July 6 to use IL-6 blockers, specifically in patients with severe/critical disease [31]. This recommendation was maintained by the NIH guidelines with a specific indication in favor of TCZ plus steroids [51], confirming the setting of patients with severe COVID-19, and was in line with the recommendation of REMAP-CAP [23] in addition to RECOVERY [28], and was consistent with definitions of progressive disease and marked pro-inflammatory status based on concentrations of C-reactive protein (CRP) being higher than 75 mg/L, a threshold established in the RECOVERY study [28].

Our meta-analysis confirms the usefulness of TCZ in ameliorating the overall survival of COVID-19 patients, especially when burdened by severe (oxygen saturation < 90% on room air, respiratory rate > 30 breaths/min, signs of severe respiratory distress) or critical (acute respiratory distress syndrome, sepsis, septic shock, provision of life-sustaining therapies such as mechanical ventilation or vasopressor therapy) disease.

The strengths of our work are represented by protocol pre-registration, focusing on the most important and objective outcome (mortality), the inclusion of only RCTs published in full after peer-review and by the carrying out of TSA, a powerful tool for clinicians to assess the conclusiveness of meta-analyses that offers better control of type I (likelihood of overestimation) and type II (likelihood of underestimation) errors [52]. Taking into account only publications that appeared in peer-reviewed journals, we had the chance to compute analyses based on consolidated data, in contrast to the processes used in previous research syntheses that relied on preliminary results from pre-prints [7,53]. For instance, as to the primary outcome, early data related to the RECOVERY study described 596 events out of 2022 patients in the TCZ arm, and 694 deaths out of 2094 subjects in the comparator group [28]. The final results were 621/2022 and 729/2094, respectively [28]. Regarding TSA, the conventional meta-analysis demonstrated statistical significance since the z-curve exceeded the monitoring boundaries to reach the so-called ‘area of benefit’, although the TSA results demonstrated that the RIS value needed to detect or reject the anticipated effect with certainty (7967) was not reached. Interestingly, the only previous evidence regarding synthesis with the TSA of RCTs reached different conclusions, describing a cumulative z-curve not crossing the boundary of benefit but crossing the one for futility due to a RIS being estimated as equal to 5622 [53]. However, the TSA conducted by Snow and colleagues [53] was based on different raw numbers of events and participants; for instance, the preliminary results of RECOVERY [28], or the 90-day mortality results of for the REMAP-CAP trial [23], as well as those of other RCTs [26,27]. Nevertheless, in our study, a sensitivity analysis that excluded the TOCIBRAS trial [30], the paramount source of heterogeneity in the leave-one-out analysis (by omitting it, the I^2^ dropped to 0%), showed that the boundary for futility was crossed. At any rate, definitive data from studies that are not still published, such as the REMDACTA trial that has enrolled patients affected by severe COVID-19 [54] and whose available results have been already included in the aforementioned prospective meta-analysis [50], are eagerly awaited since they may be conducive to a more informative TSA that can be used to establish whether additional studies are needed to confirm the usefulness of TCZ.

Of course, this study presents some limitations. In particular, owing to the publication of concurrent similar research syntheses [55,56], we did not explore secondary outcomes such as progression to mechanical ventilation, time to discharge, LOS, and safety profiles, which had already been addressed by previous works. Eventually, we did not perform meta-regression: in a preceding meta-analysis, there was no evidence of treatment effect modification by patient characteristics [57]. Nonetheless, conventional meta-analyses may be biased due to the ecological fallacy (also known as aggregation bias) since average patient characteristics are regressed against average trial outcomes; instead, individual patient characteristics should be regressed against the individual outcomes in the context of an individual patient data meta-analysis [58].

## 5. Conclusions

TCZ is one of the very few agents that has so far been found to favorably change the prognosis of patients with severe COVID-19 [59]. Nonetheless, additional RCTs are still needed to confirm this finding, upheld, beyond any reasonable doubt, by a strong biological rationale and by the data collected from completed RCTs, and to define the best schedule, in light of the different dosages administered across studies. Observational studies may have a complementary role, being instrumental in identifying adverse events and complications such as secondary bacterial infections that may develop after the usual follow-up of RCTs. Moreover, avenues for future research may be constituted by individual patient data meta-analyses and umbrella reviews. The former would allow the investigation of the effectiveness of treatment at the level of relevant patient subgroups. Granular data would permit a more precise understanding of the profile of the patients who would potentially benefit the most from the drug, besides the CRP threshold that is quite generic. The latter would allow the findings from multiple systematic reviews and meta-analyses about the review question to be compared and contrasted, and thus, would make it possible to present a wide picture of the available evidence, highlighting its consistency or potential discrepancies, in an attempt to explore and detail the underlying reasons for contradictory results.

## Figures and Tables

**Figure 1 jcm-10-04935-f001:**
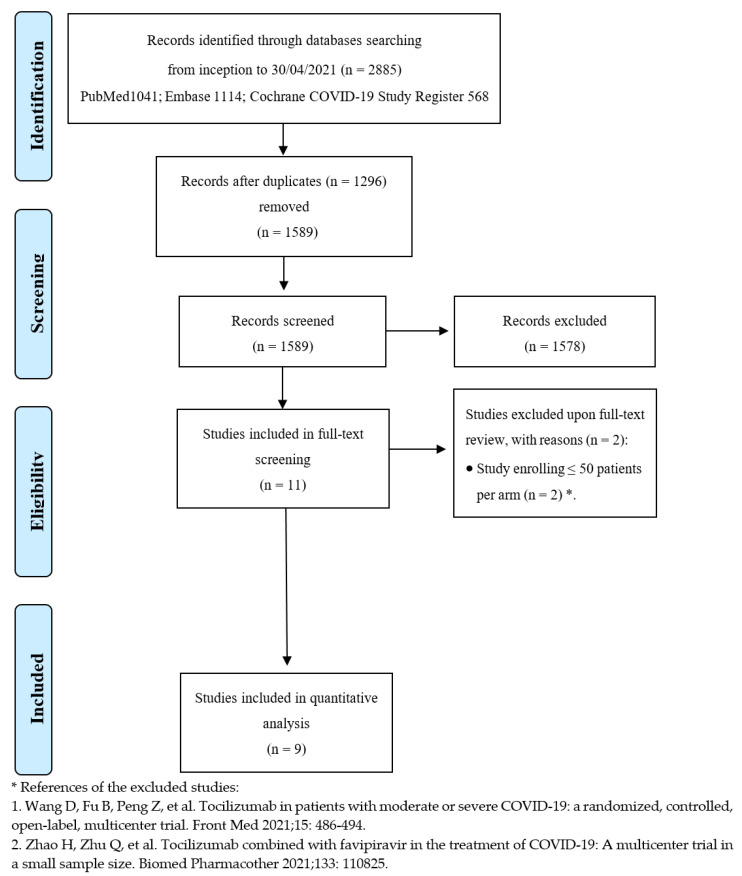
PRISMA diagram: results of the literature search and flow diagram for the selection of eligible studies.

**Figure 2 jcm-10-04935-f002:**
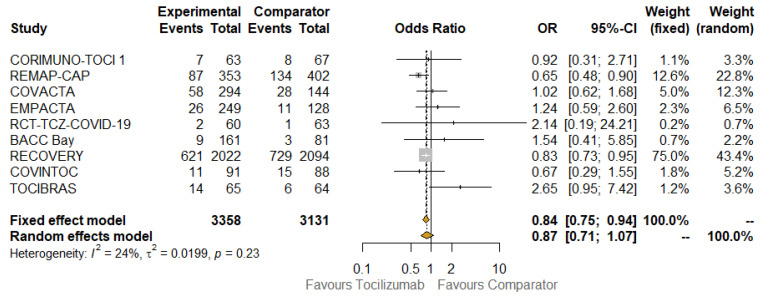
Overall meta-analysis of 28/30-day mortality. Abbreviations: OR, odds ratio; 95% CI, confidence intervals at 95%; Weight (fixed), weight of each study in a fixed-effect model; Weight (random), weight of each study in a random-effect model. Squares on the hazard ratio plot are proportional to the weight of each study; weighting is based on the inverse variance method.

**Figure 3 jcm-10-04935-f003:**
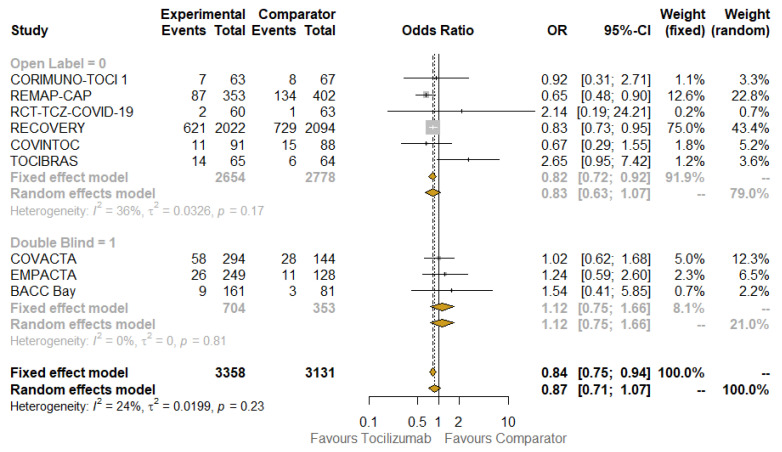
Meta-analysis of 28/30-day mortality in open-label vs. double blind studies. Abbreviations: OR: odds ratio; 95% CI, confidence intervals at 95%; Weight (fixed), weight of each study in a fixed-effect model; Weight (random), weight of each study in a random-effect model. Squares on the hazard ratio plot are proportional to the weight of each study; weighting is based on the inverse variance method.

**Figure 4 jcm-10-04935-f004:**
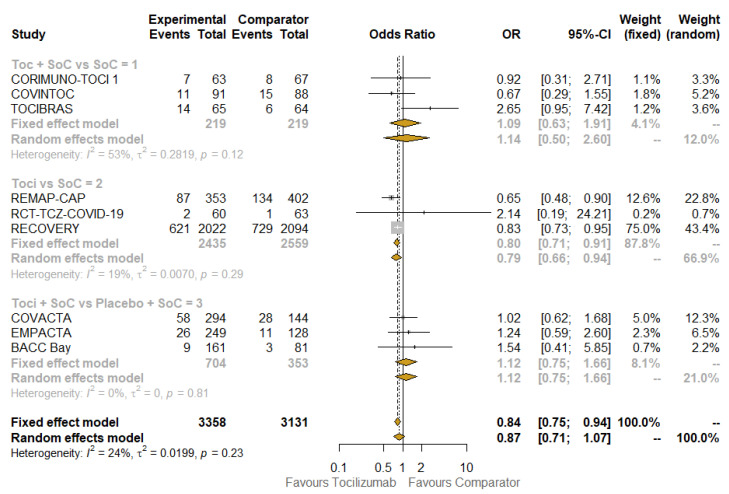
Pooled comparison of 28-day mortality according to treatment received. Abbreviations: Soc: standard of care; OR: odds ratio; 95% CI, confidence intervals at 95%; Weight (fixed), weight of each study in a fixed-effect model; Weight (random), weight of each study in a random-effect model. Squares on the hazard ratio plot are proportional to the weight of each study; weighting is based on the inverse variance method.

**Figure 5 jcm-10-04935-f005:**
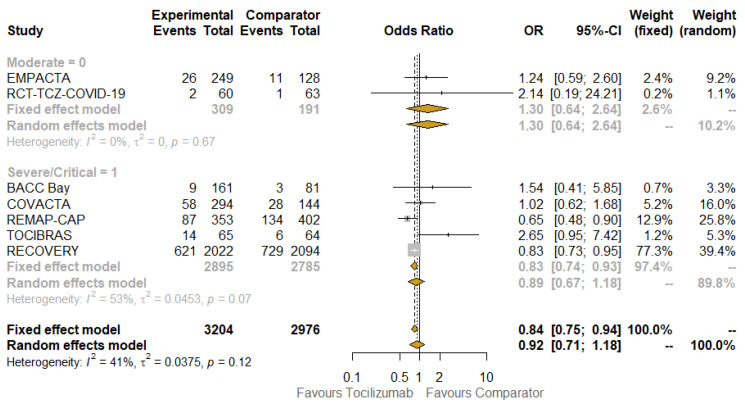
Pooled comparison of 28-day mortality according to disease status at baseline. Abbreviations: Soc: standard of care; OR: odds ratio; 95% CI, confidence intervals at 95%; Weight (fixed), weight of each study in a fixed-effect model; Weight (random), weight of each study in a random-effect model. Squares on the hazard ratio plot are proportional to the weight of each study; weighting is based on the inverse variance method.

**Figure 6 jcm-10-04935-f006:**
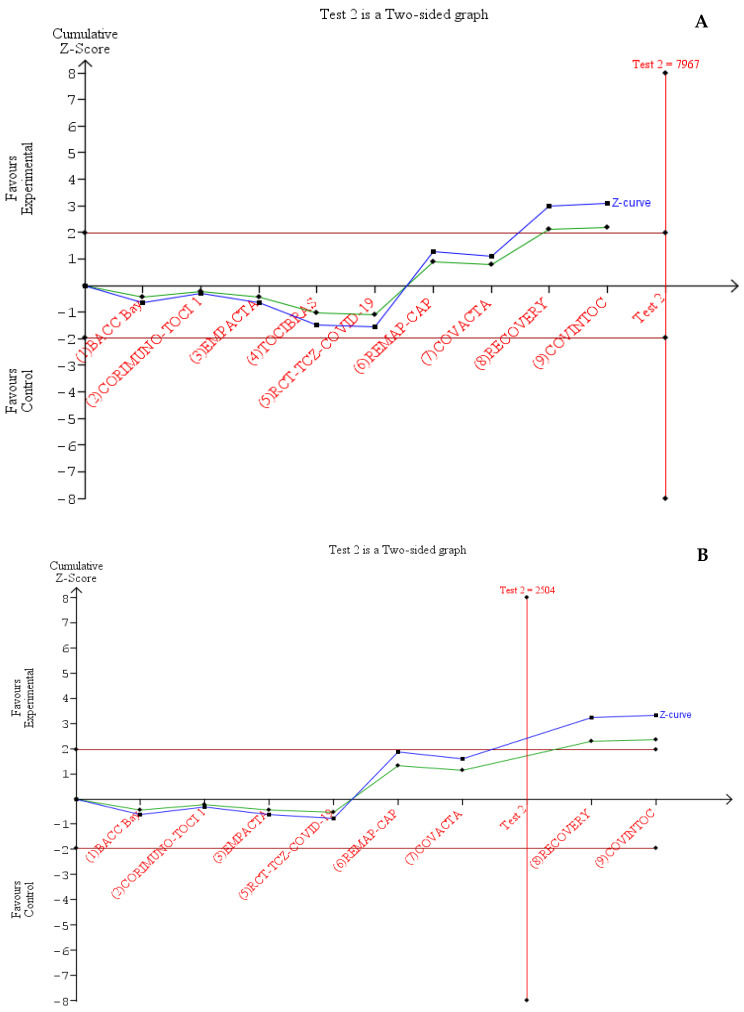
Trial Sequential analysis. (**A**) TSA of all trials included in meta-analysis. (**B**) TSA excluding the TOCIBRAS trial. The vertical red line represents the required information size to demonstrate or reject the hypothesis of a benefit from tocilizumab treatment, considering an alpha of 5% and a power of 80%. The blue line represents the cumulative z-curve, while the green line represents the cumulative z-curve (adjusted (penalized) according to law of the iterated logarithm).

**Table 1 jcm-10-04935-t001:** Characteristics of included studies (only randomized controlled trials).

First Author/Study Name/Registration Number/Reference	Design and Country	Enrolment Dates	Recruitment Window	Inclusion Criteria	Mechanical Ventilation at Baseline (%)	Treatment Group Versus Control Group (*n*)	Tocilizumab Dosing	Primary Outcome	Mortality
Hermine (CORIMUNO-TOCI 1) NCT04331808 [22]	Open label, multicenter (9 sites), France	31 March 2020 up to 18 April 2020	Within 72 h of SARS-CoV-2 diagnosis	Moderate, severe, or critical disease	0%	63 vs. 67	8 mg/Kg on day 1 (and 3 if necessary)	Scores > 5 on the WHO-CPS on day 4 and survival with no need of MV (including NIMV) at day 14	28 days
Gordon (REMAP-CAP) NCT02735707 [23]	Open label, multicenter (113 sites), international (6 countries)	19 April 2020 up to 19 November 2020	Within 24 h of ICU admission	Critical disease	29.4%	353 vs. 402	8 mg/Kg (maximum 800 mg), repeated at 12–24 h if necessary	The number of respiratory and cardiovascular organ support–free days up to day 21	In-hospital
Rosas (COVACTA) NCT04320615 [24]	Double-blind, placebo-controlled, multicenter (62 sites), international (9 countries)	3 April 2020 up to 28 May 2020	Not specified	Severe or critical disease	37.5%	294 vs. 144	8 mg/Kg (maximum 800 mg), repeated at 8–24 h if necessary	Clinical status on a 7-category ordinal scale at day 28 (1, discharged/ready for discharge; 7, death)	28 days
Salama (EMPACTA) NCT043272186 [25]	Double-blind, placebo-controlled, multicenter (69 sites), international (6 countries)	14 May 2020 up to 18 August 2020	Within 48 h of hospital admission	Severe disease	0%	249 vs. 128	8 mg/Kg (maximum 800 mg), repeated at 8–24 h if necessary	Death or MV by day 28	28 days
Salvarani (RCT-TCZ-COVID-19) NCT0436355 [26]	Open label, multicenter (24 sites), Italy	31 March 2020 up to 11 June 2020	Not specified	Severe disease	0%	60 vs. 66	8 mg/Kg (maximum 800 mg), repeated at 12 h	Occurrence of the following events, whichever came first: Admission to ICU with MV; Death (any cause); PaO_2_/FiO_2_ ratio less than 150 mmHg (confirmed within 4 h by a second examination)	30 days
Stone (BACC Bay) NCT04356937 [27]	Double-blind, placebo-controlled, multicenter (7 sites), United States	20 April 2020 up to 15 June 2020	Upon hospital admission	Severe disease	0%	161 vs. 81	8 mg/Kg as single dose	Intubation or death	28 days
Horby (RECOVERY) NCT04381936 [28]	Open label, multicenter (177 sites), United Kingdom	23 April 2020 up to 24 January 2021	Within 21 days of primary randomization	Severe and critical disease	14%	2022 vs. 2094	800 mg if weight > 90 kg; 600 mg if weight > 65 and ≤90 kg; 400 mg if weight > 40 and ≤65 kg; and 8 mg/kg if weight ≤ 40 kg); repeated at 12–24 h if necessary	All-cause mortality	28 days
Soin (COVINTOC) CTRI/2020/05/025369 [29]	Open label, multicenter (12 sites), India	30 May 2020 up to 21 August 2020	Upon hospital admission	Moderate and severe disease	5%	91 vs. 88	6 mg/Kg (maximum 480 mg), repeated within 12 h-7 days from the first dose	Proportion of patients with progression of COVID-19 from moderate to severe or from severe to death up to day 14	28 days
Veiga (TOCIBRAS) NCT04310228 [30]	Open label, multicenter (9 sites), Brazil	8 May 2020 up to 17 July 2020	Symptoms for more than 3 days	Severe and critical	16%	65 vs. 64	8 mg/Kg (maximum 800 mg) as single dose	Clinical status on a 7-category ordinal scale at day 15 (1, not admitted to hospital and with no limitation of activities; 7, death)	28 days

Legend: WHO-CPS, World Health Organization 10-point Clinical Progression Scale; MV, Mechanical ventilation; NIMV, non-invasive mechanical ventilation; ICU, intensive care unit; PaO_2_/FiO_2_ ratio, ratio of arterial oxygen partial pressure (PaO_2_) to fractional inspired oxygen (FiO_2_).

## Data Availability

The datasets generated during the current meta-analysis are available from the corresponding author upon reasonable request. All data analyzed during this meta-analysis are included in the corresponding published articles, as reported in Table 1.

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
