# Peer review of "The Use of Tocilizumab in Patients with COVID-19: A Systematic Review, Meta-Analysis and Trial Sequential Analysis of Randomized Controlled Studies"

_jcm, 2021, doi:10.3390/jcm10214935_

Round 1

Reviewer 1 Report

Dear Authors, compliments for your work. 

I have just few suggestions.

Try to shorten discussion if possible.

page 2 line 69: I suggest to cite this work as reference--> "The COVID-19 pandemic: is our medicine still evidence-based? Ir J Med Sci. 2021 Feb;190(1):11-12. doi: 10.1007/s11845-020-02258-8"

Best regards

Author Response

R1: First of all, thanks for the attention you paid to our manuscript and your favorable comments. We acknowledge that discussion is an ample part of the paper, but we feel that the current length is appropriate in order to address all the paramount point of the analysis and to reply to comments from other reviewers’ as well.

page 2 line 69: I suggest to cite this work as reference--> "The COVID-19 pandemic: is our medicine still evidence-based? Ir J Med Sci. 2021 Feb;190(1):11-12. doi: 10.1007/s11845-020-02258-8"

R2: We thank for the suggestion. The citation has been added (number 9) and bibliography has been updated accordingly.

Reviewer 2 Report

I have reviewed the systematic review, meta-analysis and trial sequential analysis of randomized controlled studies of Tocilizumab in Covid-19 patients. Before considering the work for plausible publication by the journal I have few queries to be made.

  1. Please discuss the specific exclusion criteria for n = 1578 documents
  2. Not All Randomized Trials Are Equal
  3. It is also possible that treatment-related adverse events and secondary infections will become more apparent over time, although these were rare in the studies described herein. What is the author opinion on this considering the present analysis?
  4. Moreover, tocilizumab in COVID-19 have not shown clear evidence of efficacy (for eg., Table 1, NCT043272186, 0436355, 04356937, 04381936), in contrast to observational studies. The findings do not support the routine use of tocilizumab for COVID-19 in most settings. How the authors correlate to this.
  5. Further there are numerous studies published on similar theme, how author rate their work (differences and similarities) considering following articles: 

A. Malgie J, Schoones JW, Pijls BG. Decreased Mortality in Coronavirus Disease 2019 Patients Treated with Tocilizumab: A Rapid Systematic Review and Meta-analysis of Observational Studies. Clin. Infect. Dis. 2021; 72:E742-E9. Doi: 10.1093/cid/ciaa1445

B. Tleyjeh IM, Kashour Z, Riaz M, Hassett L, Veiga VC, Kashour T. Efficacy and safety of tocilizumab in COVID-19 patients: a living systematic review and meta-analysis, first update. Clinical Microbiology and Infection 2021; 27:1076-82. Doi: 10.1016/j.cmi.2021.04.019

C. Kow CS, Hasan SS. The effect of tocilizumab on mortality in hospitalized patients with COVID-19: a meta-analysis of randomized controlled trials. European Journal of Clinical Pharmacology 2021; 77:1089-94. Doi: 10.1007/s00228-021-03087-z

D. Jiang W, Li W, Wu Q, Han Y, Zhang J, Luo T, et al. Efficacy and Safety of Tocilizumab Treatment COVID-19 Patients: A Case-Control Study and Meta-Analysis. Infectious Diseases and Therapy 2021; 10:1677-98. Doi: 10.1007/s40121-021-00483-x

E. Klopfenstein T, Gendrin V, Gerazime A, Conrozier T, Balblanc JC, Royer PY, et al. Systematic Review and Subgroup Meta-analysis of Randomized Trials to Determine Tocilizumab’s Place in COVID-19 Pneumonia. Infectious Diseases and Therapy 2021; 10:1195-213. Doi: 10.1007/s40121-021-00488-6

F. Wei Q, Lin H, Wei RG, Chen N, He F, Zou DH, et al. Tocilizumab treatment for COVID-19 patients: a systematic review and meta-analysis. Infectious Diseases of Poverty 2021; 10. Doi: 10.1186/s40249-021-00857-w

G. Snow TAC, Saleem N, Ambler G, Nastouli E, Singer M, Arulkumaran N. Tocilizumab in COVID-19: a meta-analysis, trial sequential analysis, and meta-regression of randomized-controlled trials. Intensive Care Medicine 2021; 47:641-52. Doi: 10.1007/s00134-021-06416-z

H. Lin WT, Hung SH, Lai CC, Wang CY, Chen CH. The effect of tocilizumab on COVID-19 patient mortality: A systematic review and meta-analysis of randomized controlled trials. International Immunopharmacology 2021; 96. Doi: 10.1016/j.intimp.2021.107602

I. Mahroum N, Watad A, Bridgewood C, Mansour M, Nasr A, Hussein A, et al. Systematic review and meta-analysis of tocilizumab therapy versus standard of care in over 15,000 covid-19 pneumonia patients during the first eight months of the pandemic. International Journal of Environmental Research and Public Health 2021; 18. Doi: 10.3390/ijerph18179149

J. Alkofide H, Almohaizeie A, Almuhaini S, Alotaibi B, Alkharfy KM. Tocilizumab and Systemic Corticosteroids in the Management of Patients with COVID-19: A Systematic Review and Meta-Analysis. International Journal of Infectious Diseases 2021; 110:320-9. Doi: 10.1016/j.ijid.2021.07.021

K. onti V, Corbi G, Sellitto C, Sabbatino F, Maci C, Bertini N, et al. Effect of tocilizumab in reducing the mortality rate in covid-19 patients: A systematic review with meta-analysis. Journal of Personalized Medicine 2021; 11. Doi: 10.3390/jpm11070628

Author Response

First of all, thanks for the attention you paid to our manuscript.

  1. Please discuss the specific exclusion criteria for n = 1578 documents

R1: As described by the PRISMA 2020 guidance, in the flow diagram reasons for exclusion must be provided only for papers assessed as full texts, and not in the stage of abstract/title screening (http://www.prisma-statement.org/PRISMAStatement/FlowDiagram).

  1. Not All Randomized Trials Are Equal

R2: We totally agree, and this is why we performed quality evaluation as well as sensitivity analysis to show trials whose features (e.g., low sample size) may have contributed the most to statistical and clinical heterogeneity.

  1. It is also possible that treatment-related adverse events and secondary infections will become more apparent over time, although these were rare in the studies described herein. What is the author opinion on this considering the present analysis?

R3: We take the point. We consider observational studies potentially useful to identify these complications and we added a statement in the conclusions, lines 433-435 (“Observational studies may have a complementary role, instrumental in identifying adverse events and complications such as secondary bacterial infections that may develop after the usual follow-up of RCTs”)

  1. Moreover, tocilizumab in COVID-19 have not shown clear evidence of efficacy (for eg., Table 1, NCT043272186, 0436355, 04356937, 04381936), in contrast to observational studies. The findings do not support the routine use of tocilizumab for COVID-19 in most settings. How the authors correlate to this.

R4: We agree that one size does not fit all, and tocilizumab is not efficacious in the same way in all patients. This is why we performed many analyses aimed at nailing down the category of patients that may benefit the most from the drug. Coherently with other studies, we identified severe COVID-19 as the setting wherein the drug is most effective.  About observational studies, we maintain that they may be useful to better assess the safety profile, also in the long term (please see R4), but regarding efficacy many biases usually compromise interpretation as stated in the introduction (please refer to citation 8).

  1. Further there are numerous studies published on similar theme, how author rate their work (differences and similarities) considering following articles (from A to K).

R5: We are not oblivious to the great number of systematic reviews on the topic. You gently cited 11 works but to date (19/10/21) a simple PubMed search yields more than 70 records. We humbly think that the best way to comment all these studies is to carry out an umbrella review, as suggested in the conclusions. In the discussion we focused on the most similar work, the one by Snow and colleagues, another trial sequential analysis, the only one existing in the literature on tocilizumab in COVID-19 patients; we largely comment about the difference between our work and the one by Snow.

Round 2

Reviewer 2 Report

Authors have addressed/defended most of the raised queries satisfactorily and therefore i have no hesitation in accepting the revised manuscript in the current form.